# VC-VAE: Enhancing Video VAE with Video Codec Standard for Latent Video Diffusion Model

## Abstract

Video Variational Auto-Encoders (Video VAEs) compress video data from the highly redundant pixel space into a compact latent representation, playing an important role in state-of-the-art video generation models. However, existing methods typically learn inter-frame correlations implicitly, overlooking the potential of breaking down video compression into two separate parts: keyframe encoding and inter-frame dynamic encoding, which is a fundamental design of traditional video codecs. To address this, we incorporate traditional video codec standard design into the Video VAE and introduce VC-VAE, a model that explicitly separates keyframe and inter-frame dynamic compression. We start by establishing a high-fidelity static keyframe anchor through initialization from a powerful pre-trained image VAE. Then, to explicitly model dynamic relative to this anchor, we introduce the Temporal Dynamic Difference Convolution (TDC), an operator designed to learn sparse motion residuals from inter-frame differences while maintaining a separate pathway for static content. Qualitative and quantitative experiments show that our proposed VC-VAE significantly outperforms baseline models in reconstruction quality, dynamic modelling, and training efficiency.

## 1 Introduction

Recent advances in video generation field enable the creation of high-quality videos (Wan et al., 2025; Kong et al., 2024; Polyak et al., 2024; Seawead et al., 2025; Ma et al., 2025). The progress is mainly built upon the Latent Diffusion Model (LDM) architecture (Rombach et al., 2022), which relies heavily on the Video VAE as its core component. The task of the Video VAE is to compress video data from the highly redundant pixel space into a compact latent representation, directly influencing the efficiency of the following diffusion process and the quality of the final generated videos.

Extensive researches have been conducted on enhancing Video VAEs. From the signal processing viewpoint, WF-VAE (Li et al., 2025) and Cosmos Tokenizer (Agarwal et al., 2025) aim to handle the spatio-temporal reductant in the frequency domain. By employing Haar Wavelet Transform (Procházka et al., 2011), these methods efficiently preserve the structurally-vital but temporally-redundant low-frequency content, thereby freeing up model capacity to focus on the more complex, high-frequency motion. Another line of works focus on basic architectural design. For instance, MovieGen (Polyak et al., 2024) uses a non-causal architecture to improve inter-frame modeling, whereas IV-VAE (Wu et al., 2025b) leverages a grouping mechanism to create local bidirectional dependencies, which enhances reconstruction stability while maintaining causality. However, leveraging the design principles of video codecs (Wiegand et al., 2003) to improve Video VAEs remains a sparsely explored area.

Rethinking the evolution of convolutional operators in Video VAEs uncovers an implicit parallel with the video codecs standard. As shown in Fig. 1(a), Video codecs standard(Wiegand et al., 2003) employ a Group-of-Pictures structure based on three types of frames: I-frames, which are independently encoded as self-contained anchor points; P-frames, which efficiently encode motion residuals relative to a previous frame; and B-frames, which further enhance compression by referencing frames in both past and future context. Early Video VAEs (Yu et al., 2023; Yang et al.,

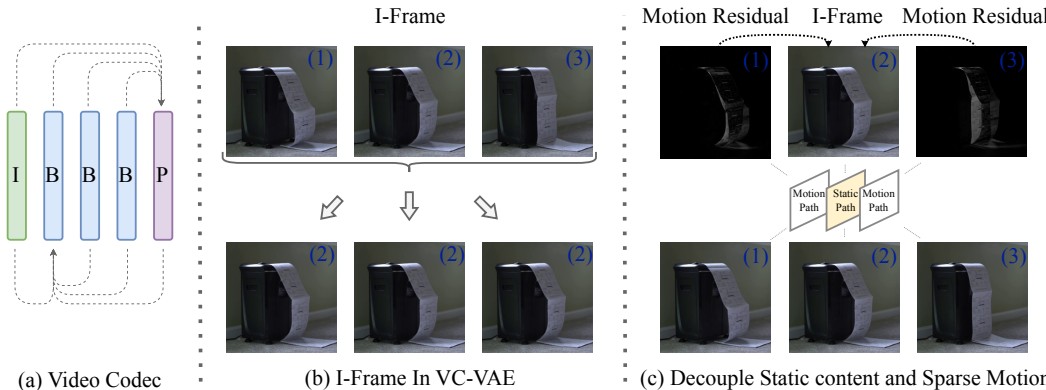

Figure 1: (a) Inter-frame dependency in Video Codec. (b) Our model first anchors the reconstruction on a single, representative I-frame. (c) Our proposed TDC explicitly learning from the decoupled sparse motion residual from the I-frame.

2024) used causal convolution, establishing an inter-frame dependency similar to P-frame prediction where the current frame is reconstructed using only past and current context. Subsequent work, such as IV-VAE (Wu et al., 2025b), introduced group causal convolution, which creates a B-frame-like dependency by utilizing both past and future context for more stable reconstruction. These architectures implicitly reflect the dependency structure of P and B frames; However, they treat static content and dynamic motion as an entangled learning target, overlooking the essential operational principle behind them. Video codecs leverage temporal redundancy through the I-frame prediction and motion residual mechanism, enabling the explicit decoupling of static content and dynamic motion. By isolating complex, sparse motion from the redundant static background, this separation offers a fundamentally more efficient approach to video modeling.

Inspired by the above insights, our objective is to enhance both reconstruction quality and training efficiency by incorporating the above design principles of video codec into Video VAE. We start by establishing an I-frame prior within the Video VAE, initializing our model from a pre-trained image VAE (Labs, 2024) to endow our Video VAE with a strong static content prior. As shown in Fig. 1(b), the model can generate high-fidelity, single-frame reconstructions at the initial state, providing a stable foundation for learning dynamic motion. We then introduce the Temporal Dynamic Difference Convolution (TDC) to enable decoupling at the operator level. As shown in Fig. 1(c), it creates a specific pathway to preserve the static content from the I-frame prior, while establishing a parallel pathway to learn sparse motion residuals from explicit feature differences between frames. Extensive experiments validate the efficiency of our video codec-inspired learning paradigm. By explicitly separating the learning of dynamic sparse motion from static content, our model achieves superior reconstruction performance while significantly reducing training overhead.

In summary, the core contributions of this paper are as follows: (1) We introduce VC-VAE, a novel architecture inspired by the video codec standard. It explicitly models the video sequence by separating it into the static I-frame and dynamic motion residuals. To the best of our knowledge, VC-VAE is the first work to successfully incorporate this explicit content-motion decoupling paradigm from video codecs into the Video VAE framework. (2) To incorporate this explicit modeling of inter-frame motion residuals into the convolutional Video VAE architecture, we introduce the Temporal Dynamic Difference Convolution (TDC), a novel operator designed for this purpose. (3) Extensive experiments demonstrate the SOTA video reconstruction capabilities of the proposed VC-VAE.

## 2 RELATED WORK

### 2.1 VARIATIONAL AUTOENCODER

The Variational Autoencoder (VAE) (Kingma & Welling, 2013) is a foundational paradigm in generative modeling, designed to capture complex probability distributions within high-dimensional data. VAEs' development of both continuous (Rombach et al., 2022; Chen et al., 2024a) and discrete latent representations (Esser et al., 2021) has established them as a versatile component in modern

generative pipelines, functioning as visual compressors for latent diffusion models and as visual to-kenizers for autoregressive systems. Contemporary research on continuous VAEs is largely geared towards enhancing their efficiency as the first stage of latent diffusion systems, pursuing higher compression rates (Chen et al., 2024a), faster throughput (Zou et al., 2025), and ensuring better compatibility with latent diffusion systems (Yao et al., 2025; Kouzelis et al., 2025; Skorokhodov et al., 2025).

## 2.2 VIDEO VAE

Early video generation systems employ SVD-VAE (Blattmann et al., 2023), which does not perform temporal compression, necessitating temporal interaction layers to ensure temporal consistency. CV-VAE (Zhao et al., 2024) introduced temporal compression into the Video VAE framework, while OD-VAE (Chen et al., 2024b) and CogVideoX-VAE (Yang et al., 2024) incorporated the temporal causal structure (Yu et al., 2023) for temporal tiling. WF-VAE (Li et al., 2025) explored wavelet transforms to efficiently reduce the expensive representation dimensionality of high resolutions. IV-VAE (Wu et al., 2025b) proposed the dual-stream architecture and Group Causal Convolutions for robust inter-frame information interaction. To reduce the computational cost in Diffusion Trans-formers (Peebles & Xie, 2023), H3AE (Wu et al., 2025c) and Reducio VAE (Tian et al., 2024) aim for extremely high compression rates while maintaining the reconstruction capability. However, existing Video VAEs merely exploit redundancy through temporal compression, without explicitly considering the inherent redundant nature of video data. Inspired by video codecs standard, we incorporate the principle of separating static content from dynamic motion into our Video VAE design.

## 3 METHOD

This section details our approach to integrating video codec principles into our VC-VAE. The overall model architecture is illustrated in Fig. 2. First, we introduce the existing convolution operators in Sec. 3.1. Then, we build a reliable keyframe prior by initializing the model with a pre-trained image VAE in Sec. 3.2. Subsequently, to explicitly model temporal dynamics, we introduce the Temporal Dynamic Difference Convolution in Sec. 3.3. Finally, we describe additional refinements for consistent reconstruction in Sec. 3.4.

### 3.1 PRELIMINARY

The core task of Video VAE is to learn a compressed yet informative latent representation for video data. Traditional video codec standards, designed explicitly for reducing the spatio-temporal re-dundancy in videos, offer invaluable insights. Motivated by this, our preliminary section revisit the fundamental operators of Video VAEs, such as Causal and Group Causal Convolutions, from the perspective of video codec principles.

#### 3.1.1 CAUSAL CONVOLUTION

To achieve unified encoding of images and videos in a VideoVAE, Causal 3D Conv (Yu et al., 2023; Yang et al., 2024) is widely adopted, which is achieved by applying asymmetric padding only at the beginning of the temporal dimension, so the receptive field at time step $t$ only covers current and past frames. Formally, for input sequence $\{X_0, ..., X_{T-1}\}$, the output $Y_t$ at time step $t$ is:

$$Y_t = CausalConv(X_t, X_{t-1}, \ldots, X_{t-k+1}) \quad (1)$$

where *CausalConv* represents a 3D convolution with temporal kernel size $k$. However, from a video codec perspective, the unidirectional dependency resembles P-frame prediction, leading to an asym-metric temporal receptive field. By only looking backwards in the temporal dimension, the model lacks future context, resulting in an information imbalance that limits its ability to model inter-frame dynamics effectively.

#### 3.1.2 GROUP CAUSAL CONVOLUTION

Group Causal Convolution (GCConv) (Wu et al., 2025b) mitigates this issue by introducing local bi-directional interactions, fostering stronger local dependencies by segmenting the video sequence

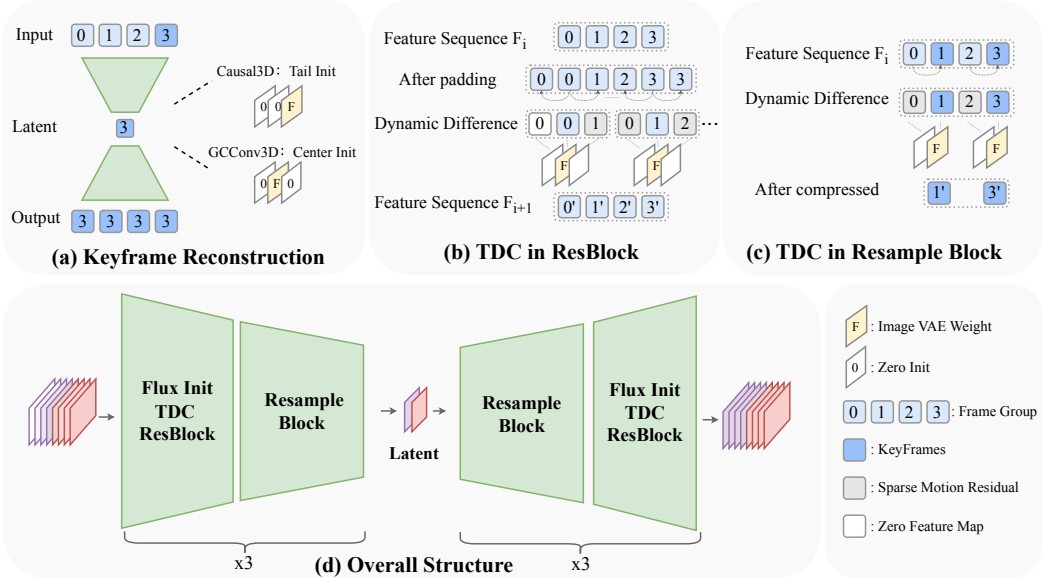

Figure 2: (a). Initialization with the Flux VAE enables the Video VAE to reconstruct keyframes. (b)-(c). Implementation of the Temporal Dynamic Difference Convolution operator within different blocks of the Video VAE. (d). Overall structure of the proposed Video VAE.

into local uncausal frame groups. Formally, for an input frame group $G_i = \{X_0, ..., X_{T-1}\}$, the output group $Y_i$ is computed as:

$$Y_i = Conv(\text{Concat}[G_{i-1}[-1], G_i]) \tag{2}$$

where $G_{i-1}[-1]$ is the last frame from the past group to maintain global causality. GCConv's advantage manifests in the superiority of bi-directional frames over predictive frames in video codec, as it allows the frame group to be encoded with richer, bi-directional context. The success of GCConv highlights the benefits of borrowing principles from video codecs. Motivated by this, the following section further instantiates the principle of video codec within the Video VAE architecture.

## 3.2 KEYFRAME RECONSTRUCTION

While basic operators effectively mirror the inter-frame interaction of a video codec, an efficient encoding scheme also depends on high-quality Intra-encoded keyframes, which are self-contained, independently compressed frames serving as robust anchors for subsequent predictions. To integrate this essential feature into our Video VAE, we initialize its weights with a powerful, pre-trained image VAE, giving our model a strong spatial prior before it begins to learn temporal dynamics. We start by inflating the 2D kernels of the image VAE into 3D kernels to enable the model to act as a high-fidelity frame-wise encoder or decoder, employing Tail Initialization (Chen et al., 2024b) for Causal Convolution and Center Initialization (Carreira & Zisserman, 2017) for Group Causal Convolution, as shown in Fig. 2(a). To leverage the temporal redundancy inherent in video data, we configure Temporal Layers for Keyframe Selection and Broadcast. Following IVVAE (Wu et al., 2025b), we utilize a convolutional layer with a temporal kernel size $k_t = 2$ and stride $s_t = 2$, denoted as $W_{down} = [C, C, k_t]$, where $C$ represents the number of input and output channels. We then initialize it as follows:

$$W_{\text{down}}[:, :, k_t - 1] = \mathbf{I}. \tag{3}$$

where $\mathbf{I} \in \mathbb{R}^{C \times C}$ denotes an identity matrix. For temporal upsampling, the operator is implemented by a convolution that doubles the channel from $C$ to $2 * C$, followed by a PixelShuffle operation to expand the temporal dimension, denoted as $W_{up} = [2 * C, C]$. Then we initial it as:

$$W_{\text{up}}[2 * C, C] = \text{concat}(\mathbf{I}, \mathbf{I}). \tag{4}$$

As shown in Fig. 2 (a), when processing an input sequence consisting of frames $\{0, 1, 2, 3\}$, the output of the encoder corresponds to the latent representation of frame $\{3\}$, which is selected

as the keyframe. Meanwhile, the decoder broadcasts the latent representation of {3} across the entire sequence, resulting in {3, 3, 3, 3}. At the start of training, the model's behavior is to select a keyframe from a video clip, encoding and decoding it using the powerful inherited image VAE and subsequently broadcasting this static frame to reconstruct the entire clip. Consequently, the learning objective is simplified: instead of learning complex visual features from scratch, the model only needs to learn the temporal dynamics required to animate the static keyframe into a dynamic sequence. This drastically reduces training costs and provides a more efficient learning task.

### 3.3 TEMPORAL DYNAMIC DIFFERENCE CONVOLUTION

With a strong spatial prior established by our keyframe-centered initialization, the model's primary learning objective shifts from reconstructing entire frames to capturing sparse temporal dynamics. To explicitly enforce this learning paradigm, we introduce the Temporal Dynamic Difference Convolution (TDC), an operator designed to follow the efficient prediction-residual design principle of video codecs. Unlike standard convolutions that process entangled features, TDC operates on the explicit feature differences between frames. It creates two distinct pathways: one that preserves the static content from the keyframe prior, and a parallel one dedicated to learning a sparse motion residual. As illustrated in Fig. 2, we incorporate specialized variants of TDC into the different blocks of our Video VAE to implement this decoupled strategy.

TDC decomposes the features within its receptive field into a static anchor frame and a set of dynamic difference features, forcing the network to learn from motion explicitly. When TDC is integrated into a Group Causal Convolution Resblock, the central frame $X_t$ of the group naturally serves as the static anchor. The TDC operator is formulated to be equivalent to a standard convolution while the kernel is explicitly re-parameterized for separate modeling, with no additional parameters or computational overhead:

$$Y_t = W_a * X_t + Wp * (X_{t-1} - X_t) + W_f * (X_{t+1} - X_t) \tag{5}$$

Here, $W_a$, $W_p$, $W_f$ are the temporal slices of the 3D kernel. The $W_a * X_t$ term, backed by our image VAE initialization, preserves the anchor frame's spatial information, while the other terms are forced to learn from the motion residuals between the anchor and its neighbors, as shown in Fig. 2(b).

We further apply TDC to model inter-frame interactions. Our temporal interaction module operates on two distinct streams: the anchor $X_a$ itself and the motion residuals $(X_p - X_a)$. The final output is a learned combination of these two streams:

$$Y_{output} = W_a * X_a + W_p * (X_p - X_a) \tag{6}$$

In this way, the model is explicitly guided to distinguish between static scene structure and temporal dynamics, as shown in Fig. 2(c). For Causal Convolutions, the anchor is always the current frame $X_t$. The operation thus learns from the differences between past frames and the current frame:

$$Y_t = W_t * X_t + W_{t-1} * (X_{t-1} - X_t) + W_{t-2} * (X_{t-2} - X_t) \tag{7}$$

This enforces the model encodes the current frame's content while explicitly learning from the historical residuals within its receptive field. This design ensures that at the start of training, its output is mathematically equivalent to that of the model initialized with the Image VAE, and it preserves the static reconstruction capability inherited from the image VAE throughout training, leading to enhanced stability and superior results.

### 3.4 OTHER IMPLEMENTATION REFINEMENT

We identified a common issue in current causal Video VAEs(Wan et al., 2025; Wu et al., 2025b): the first frame of a video sequence often has poorer reconstruction quality than later frames. This problem likely stems from the widely used Separate First-Frame Processing approach, which creates an architectural asymmetry since the first frame is processed differently due to the lack of prior temporal context. As illustrated in Fig. 4, this can cause a noticeable performance gap between first frame and the following sequence. To prevent our model from suffering from this flaw, we implement an In-Sequence First-Frame Processing pipeline that removes the special treatment of the first frame. This approach also unifies the processing of single images and video sequences by treating all inputs equally: a single image is turned into a minimal pseudo-video by repeating the frame. Consequently, all frames are processing with the same encoding-decoding process, eliminating the reconstruction inconsistencies caused by the Separate First-Frame Processing.

Figure 3: Reconstruction results of different methods for a video sequence.

Table 1: Video reconstruction results on WebVid-test and UCF-101 val. ↓ indicates lower is better, ↑ indicates higher is better.

| Method | FCR | Chn | WebVid-10M | | | | UCF-101 | | | |
|--------|-----|-----|------|------|--------|------|------|------|--------|------|
| | | | FVD↓ | PSNR↑ | LPIPS↓ | SSIM↑ | FVD↓ | PSNR↑ | LPIPS↓ | SSIM↑ |
| CogvideoX | 4*8*8 | 16 | 41.70 | 35.23 | 0.04125 | 0.9394 | 50.47 | 37.61 | 0.03292 | 0.9605 |
| Wanx2.1 | 4*8*8 | 16 | 42.40 | 35.40 | 0.03724 | 0.9385 | 43.26 | 38.19 | 0.02971 | 0.9644 |
| IVVAE | 4*8*8 | 16 | 42.80 | 35.21 | 0.04301 | 0.9343 | 40.68 | 38.48 | 0.02853 | 0.9660 |
| WFVAE | 4*8*8 | 16 | 36.86 | 35.60 | 0.03778 | 0.9394 | 42.73 | 38.31 | 0.03036 | 0.9645 |
| Hunyuan | 4*8*8 | 16 | **36.23** | 35.84 | 0.03452 | 0.9425 | 37.07 | 38.80 | 0.02757 | 0.9671 |
| VC-VAE | 4*8*8 | 16 | 36.80 | **36.04** | **0.03315** | **0.9459** | **36.30** | **38.97** | **0.02747** | **0.9684** |
| WFVAE | 8*8*8 | 32 | 44.13 | 35.86 | 0.03735 | 0.9396 | 53.11 | 37.95 | 0.03307 | 0.9627 |
| VC-VAE | 8*8*8 | 32 | **36.63** | **36.40** | **0.03298** | **0.9485** | **38.47** | **38.85** | **0.02854** | **0.9672** |

## 4 EXPERIMENTS

### 4.1 EXPERIMENTAL SETUP

**Training details.** We train our Video VAE on the Kinetics-600 dataset (Carreira et al., 2018) using a two-stage approach. We first pretrain the model for 500k steps, focusing on learning fundamental spatio-temporal dynamics from a large volume of short-duration, low-resolution videos, optimised with a combination of reconstruction losses (L1, LPIPS (Zhang et al., 2018)) and KL regularization (Kingma & Welling, 2013). This is followed by a 50k steps fine-tuning stage on a diverse mix of videos with varying resolutions, frame rates, and durations. In this stage, we incorporate an adversarial loss from a 3D discriminator to improve high-frequency details and strengthen temporal extrapolation capabilities. As shown in Table 2, we compare with the total training steps required by state-of-the-art models. Notably, our total training compute is considerably lower than that of state-of-the-art models, highlighting the training efficiency of our proposed approach.

Table 2: Comparison of total training computation.

| Method | Training steps |
|--------|----------------|
| WFVAE | 1200k |
| IVVAE | 1000k |
| VC-VAE | 550k |

For the 4×8×8 compression version, which downsamples the video by a factor of 4 in the temporal dimension and 8 in each spatial dimension, we use 16 channels identical to most of the SOTA Video VAEs, and utilize Flux VAE (Labs, 2024) for initialization. For the 8×8×8 compression version, we employ a 32-channel implementation, where the first half is initialized from Flux VAE and the remaining half is zero-initialized. This strategy ensures that, after increasing the number of channels, our model begins with an initial image reconstruction capability identical to Flux VAE.

Table 3: Results on TokBench Video. T-ACC, T-NED measure the accuracy of text reconstruction, while F-Sim measure the similarity of face reconstruction. ↑ indicates higher is better.

| Method | FCR | T-ACC ↑ | | | | T-NED ↑ | | | | F-Sim ↑ | | | |
|---|---|---|---|---|---|---|---|---|---|---|---|---|---|
| | | small | medium | large | mean | small | medium | large | mean | small | medium | large | mean |
| *Resolution: 256x* | | | | | | | | | | | | | |
| CogvideoX | 4*8*8 | 24.80 | 72.47 | 86.34 | 61.21 | 43.06 | 82.29 | 92.41 | 72.59 | 0.58 | 0.78 | 0.91 | 0.76 |
| Wanx2.1 | 4*8*8 | 17.88 | 69.52 | 87.56 | 58.32 | 37.27 | 81.04 | 93.18 | 70.50 | 0.59 | 0.79 | 0.92 | 0.77 |
| IVVAE | 4*8*8 | 18.43 | 72.08 | 89.52 | 60.01 | 38.29 | 82.49 | 94.36 | 71.72 | 0.57 | 0.79 | 0.93 | 0.76 |
| Hunyuan | 4*8*8 | 26.85 | 69.12 | 87.47 | 61.15 | 45.55 | 80.54 | 93.12 | 73.07 | 0.60 | 0.80 | 0.92 | 0.77 |
| VC-VAE | 4*8*8 | **31.81** | **77.21** | **89.70** | **66.24** | **51.21** | **85.24** | **94.39** | **76.95** | **0.64** | **0.82** | **0.93** | **0.80** |
| *Resolution: 480x* | | | | | | | | | | | | | |
| CogvideoX | 4*8*8 | 28.02 | 65.41 | 91.71 | 61.71 | 43.47 | 78.24 | 95.60 | 72.43 | 0.67 | 0.80 | 0.91 | 0.79 |
| Wanx2.1 | 4*8*8 | 18.88 | 63.42 | 92.36 | 58.22 | 35.95 | 77.43 | 96.02 | 69.80 | 0.68 | 0.82 | 0.92 | 0.81 |
| IVVAE | 4*8*8 | 20.58 | 63.89 | **92.98** | 59.15 | 37.85 | 78.07 | **96.37** | 70.76 | 0.68 | 0.82 | 0.92 | 0.80 |
| Hunyuan | 4*8*8 | 28.65 | 64.49 | 91.83 | 61.66 | 44.43 | 77.83 | 95.83 | 72.70 | 0.69 | 0.82 | 0.92 | 0.81 |
| VC-VAE | 4*8*8 | **31.24** | **72.27** | 92.72 | **65.41** | **48.11** | **82.90** | 96.24 | **75.75** | **0.72** | **0.83** | 0.92 | **0.83** |

Table 4: Effectiveness of the TDC Operator.

| Setting | PSNR↑ | SSIM↑ | LPIPS↓ |
|---|---|---|---|
| Causal Conv Baseline | 33.29 | 0.9298 | 0.04946 |
| + TDC | 33.64 | 0.9336 | 0.04599 |
| GCConv Baseline | 33.49 | 0.9323 | 0.04720 |
| + TDC (Full model) | **33.79** | **0.9357** | **0.04455** |

Table 5: Impact of the Image VAE's Performance.

| Setting | PSNR↑ | SSIM↑ | LPIPS↓ |
|---|---|---|---|
| SD3.5-VAE Init w/o TDC | 33.13 | 0.9264 | 0.05519 |
| Flux-VAE Init w/o TDC | 33.49 | 0.9323 | 0.04720 |
| SD3.5-VAE Init w TDC | 33.74 | 0.9337 | 0.04691 |
| Flux-VAE Init w TDC | **33.79** | **0.9357** | **0.04455** |

Detailed hyperparameters, including learning rates and loss weights for each stage, are provided in the supplementary material.

**Evaluation details.** We employ WebVid-10M (Bain et al., 2021) and UCF-101 (Soomro et al., 2012) to assess overall video reconstruction performance. To ensure an fair comparison, all metrics are calculated without the first frame for baseline models. We utilize FVD (Unterthiner et al., 2019), PSNR (Hore & Ziou, 2010), SSIM (Wang et al., 2004), and LPIPS (Zhang et al., 2018) to measure reconstruction quality. Furthermore, we assess the VAE's ability to reconstruct text and faces within videos using the TokBench-Video (Wu et al., 2025a) dataset, with performance evaluated by reconstruction text accuracy and face similarity (Deng et al., 2019).

## 4.2 PERFORMANCE

Our baseline models include several 4×8×8 SOTA methods: CogVideoX-VAE (Yang et al., 2024), Wan2.1-VAE (Wan et al., 2025), WFVAE (Li et al., 2025), IVVAE (Wu et al., 2025b), and Hunyuan-VAE (Kong et al., 2024). Additionally, WFVAE provides an 8×8×8 version, which serves as the baseline for our experiments on high-ratio temporal compression.

**Quantitative evaluation of reconstruction results.** We present a comprehensive quantitative evaluation of our VC-VAE against several SOTA methods in Tab. 1, with results reported on both WebVid-10M and UCF-101. Under the standard 4×8×8 compression setting, our VC-VAE achieves SOTA performance across all datasets. We also evaluate a more challenging configuration with a higher temporal compression rate, where our model continues to outperform the WFVAE baseline. These results validate the effectiveness and robustness of our proposed architecture. Furthermore, to provide a more comprehensive analysis, we test our Video VAE on the challenging TokBench-Video benchmark. This benchmark is particularly demanding as it comprises long-duration videos with complex motion and intricate text. On this benchmark, our model shows significant performance gains, especially in the reconstruction of small to medium sized text and faces, as shown in Tab. 3. This highlights our method's enhanced capability in modelling both high-frequency details and temporal dynamics.

**Qualitative evaluation of reconstruction results.** Fig. 3 demonstrates the reconstruction results for a video sequence with rapid motion. In the challenging sports scene, our method effectively captures movement while maintaining the reconstruction stability of fine details, such as text. In contrast, other methods show varying degrees of reconstruction artifacts. This offers qualitative

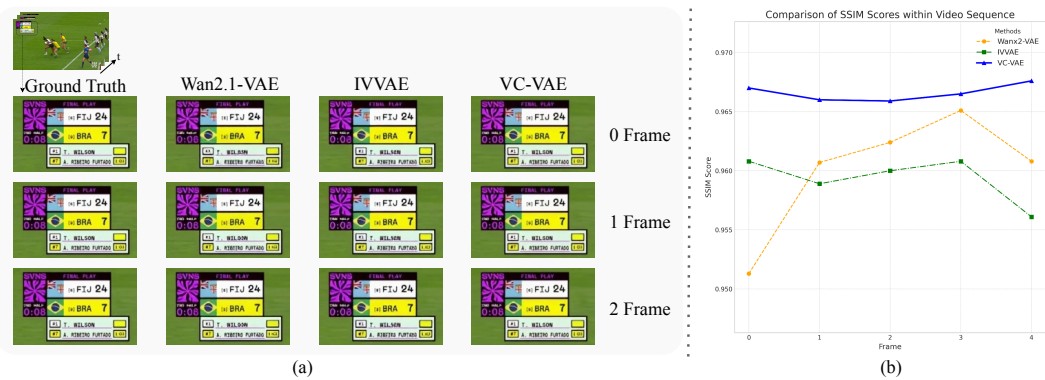

Figure 4: (a). Visualization of the video sequence's reconstruction performance. (b). SSIM score across the first five frames. Although IVVAE's SSIM score for the first frame appears normal, its visual quality degrades significantly.

evidence for the effectiveness of our proposed TDC operator in decoupling static information from dynamic motion. Additional reconstruction results are included in the supplementary material.

## 4.3 Ablation Study

In this section, we conduct a series of ablation experiments. For the ablation study, we first initialize Video VAE with Flux VAE, then train it for 100K steps on 16-frame videos with 256×256 resolution using the Kinetics-600 dataset. The test dataset comprises 2000 videos from the Kinetics-600 validation set, each with 16-frames at 256×256 resolution.

**Ablation of main components.** We conducted ablation studies to verify the effectiveness of our TDC operator. For standard Causal Convolution and Group Causal Convolution baseline, we replaced their core temporal operators with our TDC operator. As shown in the Tab. 4, the TDC operator delivers significant improvements in both PSNR and LPIPS metrics. This demonstrates that our design, inspired by video codec compression, substantially enhances the model's ability to both preserve the static details inherited from the ImageVAE and model temporal dynamics. The consistent performance gains observed across different architectures confirm the superiority and generality of our TDC operator.

**Ablation on First Frame Processing Strategy.** To validate the effectiveness of our In-Sequence First-Frame Processing strategy, we compare our implementation with the widely used Separate First-Frame Processing variant. As shown in Tab. 6, by adopting the In-Sequence First-Frame Processing, our model achieves a 1.2dB improvement in PSNR for the first frame, effectively resolving the inconsistency. More evaluation of our model's performance on single-image reconstruction is provided in the supplementary material.

Table 6: Ablation on First-Frame Processing Strategy.

| Method | PSNR↑ | SSIM↑ | LPIPS↓ |
|---|---|---|---|
| Separate | 32.15 | 0.9242 | 0.04059 |
| In-Sequence | 33.34 | 0.9346 | 0.03601 |

**Impact of the Initializing Image VAE.** To evaluate our model's sensitivity to the quality of pretrained Image VAE weights, we perform an ablation study using two image VAEs with different performance levels: a stronger Flux-VAE and a comparatively weaker SD3.5-VAE (Esser et al., 2024). As shown in Tab. 5, when training a baseline model without TDC, performance strongly depends on the quality of initialization; the model initialized with the stronger Flux-VAE notably outperforms the one starting from SD3.5-VAE. However, when our TDC operator is added, the performance gap between the two setups decreases significantly. Remarkably, adding TDC results in a more notable improvement for the model initialized with the weaker SD3.5-VAE, with a PSNR increase of 0.61 dB. This indicates that TDC not only enhances overall performance but also makes training more robust and less sensitive to the quality of initial spatial priors.

## 4.4 Visualization

**Visulization about TDC.** To intuitively understand how our Temporal Dynamic Difference Convolution operator explicitly separates the sparse motion residual from the static content, we perform

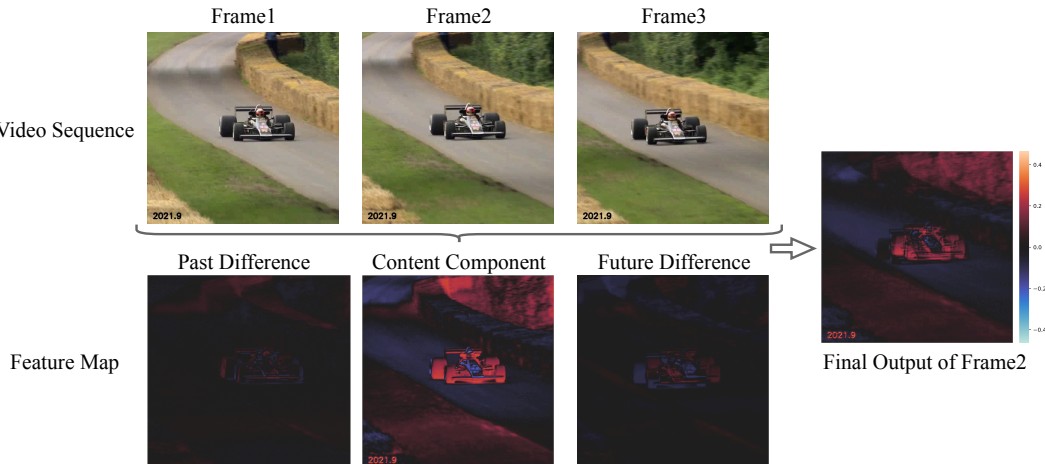

Figure 5: Visulization of the proposed TDC operator.

visualization of its internal components. As shown in Fig. 5, we use three frames from a video sequence with significant motion to clearly illustrate the effect. The anchor component focuses solely on modelling the intra-frame spatial information of the current frame, successfully encoding the representation of static structures, including the car's appearance, the background, and the watermark, while the other components, derived from the temporal differences, isolate inter-frame temporal changes. Static regions such as the watermark and stationary background elements exhibit near-zero activation. Conversely, activations are concentrated around the moving object, effectively encoding its displacement between frames. These difference maps can be interpreted as learned, high-level motion representations, highlighting where and how the scene changes over time. The final output synthesises these components by combining the rich spatial details from the content component with the precise motion information from the difference components, demonstrating that our TDC operator does not merely mix temporal information but explicitly disentangles the representation of content from its dynamics, enabling a more efficient and interpretable modelling of video.

**Visulization of First Frame Performance.** As discussed in the method section, causal Video VAEs often suffer from inconsistent reconstruction quality, particularly for the first frame. Fig. 4 provides a clear qualitative demonstration of this phenomenon. In the 0 Frame column, it is evident that baseline models like Wan2.1-VAE and IVVAE exhibit significant degradation. This quality drop appears confined to the initial frame, as their performance on subsequent frames is visibly better, highlighting the performance discrepancy stemming from architectural asymmetry. In contrast, our VC-VAE maintains a consistently high level of reconstruction fidelity across all frames. This robustness is a direct result of our In-Sequence First-Frame Processing strategy, which eliminates the performance degradation of the initial frame. This visualization confirms that our approach effectively resolves this common phenomenon in causal video vaes and ensures stable, high-quality performance throughout the entire video sequence.

## 5 CONCLUSION

Inspired by video codecs, which use keyframes to encode static content and motion residuals for inter-frame dynamics, we propose VC-VAE, a novel architecture that integrates this principle of decomposition into the design of Video VAEs. First, by initializing with a high-quality ImageVAE, we endow the model with strong keyframe reconstruction capabilities, ensuring the quality of its static spatial priors. Second, we design the Temporal Dynamic Difference Convolution operator, which explicitly computes temporal differences within convolution, successfully decoupling static spatial content from dynamic temporal motion information. The TDC operator significantly boosts the model's performance without introducing any additional parameters or computational costs, particularly enhancing its ability to model high-frequency motion details. Extensive experiments show that the proposed VC-VAE achieves SOTA results in video reconstruction.

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

# A APPENDIX

## LLM USAGE STATEMENT

In accordance with the ICLR 2026 policy on LLM usage, we disclose that large language models were used as an assistive tool in the preparation of this paper. Specifically, we utilized LLMs for: (1) refining grammar, sentence structure, and improving the readability of the text; and (2) generating LaTeX source code for some of the tables. We emphasize that all content suggested or generated by the LLM was critically reviewed, manually verified, and edited by the authors to ensure accuracy and alignment with our research. The authors are fully responsible for all claims, results, and the final text of this submission.

## REPRODUCIBLILITY STATEMENT

To ensure the reproducibility of our work, we will release our complete source code and all pre-trained model weights upon publication. Our implementation, based on PyTorch and building upon publicly available codebases, is designed to be clear and self-contained. Our experiments rely on standard public datasets (e.g., Kinetics-600, UCF-101, Web-Vid) for evaluation, and while some training involved an internal dataset that cannot be released, all key results and ablation studies presented in this paper are on public data, allowing for independent verification of our claims. Comprehensive details regarding the experimental setup are provided in the Appendix, including specific hyperparameter settings, optimizer configurations, data processing pipelines, and training procedures for each model. Furthermore, all models were trained on commercially available H20 GPUs, which can lower the barrier for reproduction without requiring specialized, high-end computational resources.

## A.1 TRAINING DETAILS

In the supplementary material, we provide further details on our training hyperparameters. Specifically, the data configurations and loss hyperparameters for each training stage are summarized in Tab. 7. During the stage I, the training is driven primarily by the L1 loss, with the GAN loss excluded. In the Stage II, we introduce GAN loss to specifically target and refine high-frequency details under more complex data.

Table 7: Training hyperparameters across two stages.

| Parameter | Stage I (500k step) | Stage II (50k step) |
|---|---|---|
| Learning Rate | 1e-5 | 1e-5 |
| Total Batch Size | 8 | 8 |
| Perceptual(LPIPS) Weight | 0.1 | 0.2 |
| GAN Weight | 0 | 0.02 |
| KL Weight | 1e-8 | 1e-8 |
| Resolution | 256 | 128, 256, 512 |
| FPS | 24 | 8, 12, 15, 24 |
| Num Frames | 16 | 8, 40, 120 |
| EMA Decay | 0.999 | 0.999 |

## A.2 IMAGE RECONSTRUCTION CAPABILITY

We demonstrate the superiority of our In-Sequence First-Frame Processing over Separate First-Frame Processing designs through two key experiments on the ImageNet validation set (Tab. 9). First, we evaluated VC-VAE's image reconstruction capability. It achieves state-of-the-art performance compared to both Image and Video VAEs, confirming that our model excels at processing static images. Second, we designed an experiment to reveal the inherent weakness of the Separate First-Frame Processing. We tested the Wanx2.1 VAE with a "pseudo-video" created by repeating a single image, denoted as Wanx2.1*. The results show a significant drop in quality compared to its

Table 9: Image reconstruction results on ImageNet-val. ↓ indicates lower is better, ↑ indicates higher is better. *denotes utilizing video frames to represent an image.

| Method | 256×256 | | | 512×512 | | | 1024×1024 | | |
|---|---|---|---|---|---|---|---|---|---|
| | PSNR↑ | SSIM↑ | LPIPS↓ | PSNR↑ | SSIM↑ | LPIPS↓ | PSNR↑ | SSIM↑ | LPIPS↓ |
| SD3.5 VAE | 31.29 | 0.877 | 0.060 | 33.54 | 0.910 | 0.060 | 37.61 | 0.971 | 0.034 |
| Flux VAE | 32.87 | 0.911 | 0.044 | 35.40 | 0.939 | 0.042 | 39.84 | 0.982 | 0.023 |
| Wanx2.1 | 31.34 | 0.879 | 0.058 | 34.25 | 0.913 | 0.056 | 40.96 | 0.973 | 0.032 |
| Wanx2.1* | 33.03 | 0.915 | **0.037** | 35.96 | 0.942 | **0.036** | 42.43 | 0.981 | **0.022** |
| Hunyuan | 33.33 | 0.910 | 0.054 | 36.03 | 0.935 | 0.053 | 43.15 | 0.982 | 0.026 |
| VC-VAE | **33.42** | **0.919** | 0.048 | **36.37** | **0.944** | 0.046 | **43.65** | **0.984** | 0.024 |

dedicated first-frame output. This performance discrepancy highlights a key limitation of separate First-Frame Processing, thus validating our design choice.

## A.3 MORE ABLATION STUDY

To further validate the effectiveness of the TDC operator, we conduct an ablation study in the supplementary materials on models with higher temporal compression rates. As shown in Tab. 8, our method yields more significant per-formance gains as the compression rate increases. This result demonstrates the effectiveness of our TDC operator in modeling temporal dynamics.

Table 8: Ablation on TDC operator.

| Method | FCR | PSNR↑ | SSIM↑ | LPIPS↓ |
|---|---|---|---|---|
| GCConv w/o TDC | 8*8*8 | 31.11 | 0.9018 | 0.07641 |
| GCConv w TDC | 8*8*8 | 32.01 | 0.9122 | 0.07099 |

## A.4 MORE RECONSTRUCTION VISULIZATION

As shown in Fig. 6, we qualitatively evaluate the reconstruction performance under two challenging scenarios: a scene with rapid motion (top rows) and another with intricate textures (bottom row). In the high-motion case, as the scoreboard scrolls down, most competing methods suffer from severe artifacts, resulting in blurred and illegible text. In contrast, our VC-VAE maintains the sharpness and clarity of the text details despite the fast movement. Furthermore, for the fine-grained texture scene, our model achieves a visually superior reconstruction of the boats, outperforming the competing methods.

## A.5 GENERATION VISULIZATION

Following previous work Wu et al. (2025b); Chen et al. (2024b), we demonstrate the video gener-ation capability of our Video VAE. We train an unconditional latent video diffusion model, using Latte-XL (Ma et al., 2024) as the generative backbone, on the SkyTimelapse (Xiong et al., 2018) dataset. The qualitative results are visualized in Fig. 7.

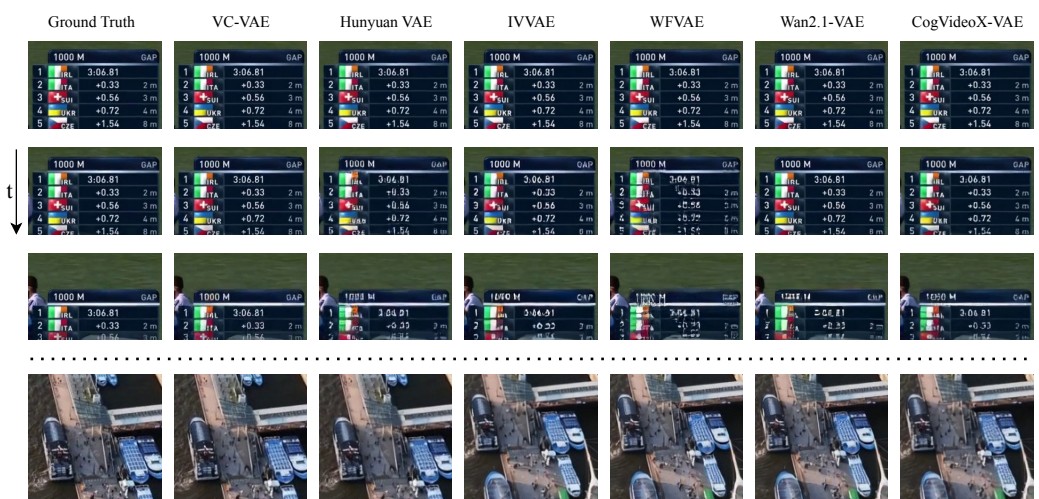

Figure 6: Reconstruction results of different methods.

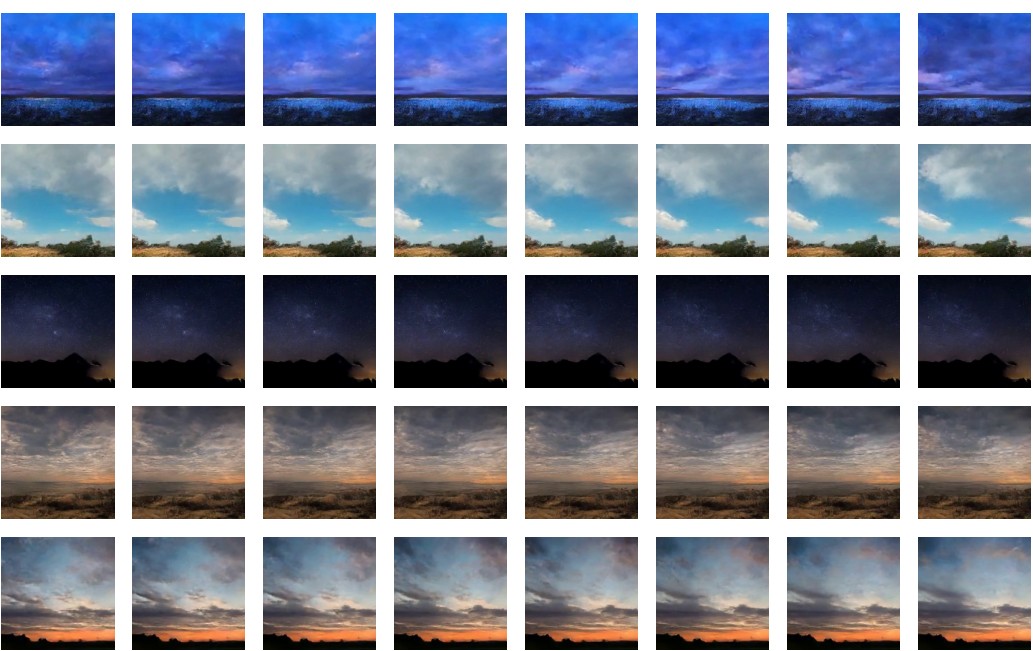

Figure 7: Visualization of the Gererative result on SkyTimeLapse.

