# OpenReview forum: "VC-VAE: Enhancing Video VAE with Video Codec Standard for Latent Video Diffusion Model"
_ICLR.cc/2026/Conference — ICLR 2026 Conference Withdrawn Submission_

### Official Review · Reviewer_zj9n · 2025-10-29

**Soundness:** 3
**Presentation:** 4
**Contribution:** 3
**Rating:** 4
**Confidence:** 5

**Summary:**

This paper proposes a new video VAE for video generation by incorporating design principles from traditional video codec standards into the VAE architecture. It initializes the model using pretrained weights from an image VAE and employs Temporal Dynamic Difference Convolution (TDC) to capture temporal dynamics. Evaluations on several benchmarks demonstrate that the proposed method outperforms existing video VAEs.

**Strengths:**

+ Integrates traditional video codec design principles into the VAE framework.
+ Carefully initializes the model with pretrained image VAE weights.
+ Achieves superior performance over other video VAEs across multiple benchmarks.

**Weaknesses:**

- The performance improvements on WebVid-10M and UCF-101 appear marginal compared to state-of-the-art video VAEs when FCR is 4×8×8, despite strong text reconstruction results. The reason for this inconsistency is not clearly explained.
- The supplementary materials do not include video samples, making it difficult to assess temporal consistency.
- Although the title claims the VAE is designed for latent video diffusion models, there is no experiment training a full video generation model based on the proposed VAE. Thus, it remains unclear how well the VAE performs in actual video generation tasks.

**Questions:**

+ Reconstruction videos should be provided.
+ A video model needs to be trained based on the proposed video VAE.

---

### Official Review · Reviewer_maVj · 2025-10-30

**Soundness:** 2
**Presentation:** 2
**Contribution:** 2
**Rating:** 2
**Confidence:** 5

**Summary:**

Following the growing importance of compact video representations for video generation, this paper revisits video codec–inspired video VAE by designing a model that explicitly separates keyframe and inter-frame dynamic compression. Starting from a pre-trained image VAE, which is efficient, the authors introduce a Temporal Dynamic Difference Convolution (TDC) operator to learn sparse motion residuals from inter-frame differences. The quantitative results in Table 1 and Table 3 of text and human-face datasets are interesting and demonstrate strong reconstruction quality.

However, the paper does not sufficiently discuss related prior works and lacks comparisons with relevant baseline models, which weakens the positioning of the claimed contributions. In addition, key experimental results such as quantitative evaluation for video generation and validation of common issues (e.g., flickering, reconstruction consistency) are missing.

**Strengths:**

- The video codec–inspired video encoding approach is interesting and shows strong reconstruction performance demonstrating that the potential benefits of incorporating codec-inspired structures into Video VAEs.
- The observation that the first frame of a video sequence often has poorer reconstruction quality than later frames is insightful and could inspire future work on temporal consistency and keyframe modeling.

**Weaknesses:**

While the core idea and experimental observations are interesting, the paper currently lacks clear validation and sufficient discussion to substantiate its main claims. Specifically, the issues described below (regarding the overstated key claim, unclear problem motivation, and insufficient quantitative validation) limit the overall strength of the contribution.

1. **Key claim is somewhat overstated**
- The paper claims that leveraging video codec design principles to improve Video VAEs remains a sparsely explored area. However, in the video compression domain, many recent works have already designed autoencoder-based architectures inspired by video codecs (e.g., [1], [2]). While most VAE-based video generative models focus on compression without explicitly disentangling keyframes and motion as in codecs, the authors should clarify how their approach fundamentally differs from these prior works. Please discuss these differences explicitly in the paper.

- Moreover, the idea of separating static content and dynamic motion has already been explored in prior works such as [3] and [4]. Although these methods may not explicitly follow codec structures, the concept of content–motion disentanglement is not new. The paper should discuss and compare against these methods in more detail to clarify its novelty.

2. **Unclear motivation for the implicit modeling problem**
- The paper claims that implicit modeling is problematic but does not provide specific failure cases or quantitative evidence to support this argument. Some experimental validation (e.g., ablations or comparisons) would help clarify what concrete problem the proposed method is solving.
- In addition, the method assumes that separating motion from a redundant static background improves efficiency, yet it is unclear how the approach performs under dynamic backgrounds, such as with camera motion, crowded scenes, or complex natural textures. Evaluating this would strengthen the motivation and validity of the proposed formulation.

3. **Experimental limitations and lack of validation**
- No quantitative evaluation for video generation. Although the method is motivated by generation via reconstruction, no FVD, KVD, or other generative quality metrics are reported. The qualitative results in the appendix also do not clearly demonstrate improvements.
- The paper mentions that we identified a common issue, but provides no formal validation protocol or statistical evidence. This remains anecdotal. Figure 4 shows only one example and lacks context: how many videos were tested, whether the phenomenon holds across the entire dataset, and whether other metrics were considered.
- Missing runtime analysis. Encoding and decoding times are not reported, making it difficult to assess the practical efficiency of the proposed approach compared to existing baselines.

[1] DVC: An End-to-end Deep Video Compression Framework, Lu et al., CVPR 2019\
[2] Neural Inter-Frame Compression for Video Coding, Djelouah et al., ICCV 2019\
[3] CMD: Efficient Video Diffusion Models via Content-Frame Motion-Latent Decomposition, Yu et al., ICLR 2024\
[4] Video Probabilistic Diffusion Models in Projected Latent Space, Yu et al., CVPR 2023\

If all of these issues are properly addressed through clearer positioning, additional experiments, and more rigorous analysis, I would be willing to raise my score.

**Questions:**

- How is the number of keyframes per clip chosen? Is it fixed or adaptively determined based on content?
- How does the model handle dynamic backgrounds where motion and content separation is ambiguous?
- Why does the ImageNet result outperform an image-only VAE baseline? What architectural or training differences explain this?

---

### Official Review · Reviewer_oa2j · 2025-10-31

**Soundness:** 3
**Presentation:** 3
**Contribution:** 2
**Rating:** 4
**Confidence:** 5

**Summary:**

The authors propose VC-VAE, a video variational autoencoders. It inherits the concepts from advanced video codecs to compress a video into two separate parts; key frame and intra frame. VC-VAE incorporates traditional codec design to explicility compress the two parts separately. It firstly use an image VAE to encode key frames, then a proposed temporal dynamic difference convolution operator is used to extract the inter-frame information. Extensive experiments show the proposed VC-VAE achieves promising performance.

**Strengths:**

1. This paper proposes a novel way to build video VAEs from a image VAE, by adding the TDC operators in resblocks and resample blocks.
2. The method is tested in two VAEs including SD3 and Flux.
3. Expriments show VC-VAE achieves the state-of-the-art reconstruction quality on video data.

**Weaknesses:**

1. The trainability of a video generator on VC-VAE is not discussed in this paper. The author should try to show VC-VAE latents are trainable for video generators (e.g, DiT). For example, after training the model for the same steps, the video generator on VC-VAE will not achieve a inferior performance than the traditional video VAEs.
2. Figure 3 can hardly tell why VC-VAE is better than the other methods. The authors can try to zoom in the differences or choose to use another example.
3. Attached videos are crucial for reviewing the paper. It helps greatly to determine how well DC-VAE is doing. However, no attachment is provided in the supplementary material.
4. A throughput comparison is missing. It can help readers understand the computation efficiency of VC-VAE.
5. To show TDC is effective in accelerating the training. The authors can provide a comparison on the training curves between w. TDC and w.o. TDC.

**Questions:**

1. In Tab.3, what initialization model is used for VC-VAE?
2. Compared to the baselines, how many parameters VC-VAE have?
3. Advanced video codec will adaptively deterimine the density of I frames depending on the complexity of the video. Do you have any idea to simulate that process in VAE video codec?

---

### Official Review · Reviewer_Gg4s · 2025-11-01

**Soundness:** 2
**Presentation:** 1
**Contribution:** 2
**Rating:** 2
**Confidence:** 4

**Summary:**

This paper introduces **VC-VAE**, a novel Video Variational Auto-Encoder architecture designed to improve video compression and reconstruction quality by incorporating principles from traditional video codecs. The model is first initialized with a powerful pre-trained image VAE to establish a high-fidelity "I-frame" or keyframe anchor. This provides a strong prior for static content. The model then uses a novel operator, the **Temporal Dynamic Difference Convolution (TDC)**, to explicitly learn sparse "motion residuals" from the differences between frames. TDC is designed with separate pathways to preserve static content while only learning the dynamic changes.

**Strengths:**

The paper's primary strength lies in its originality and significance, which are communicated with exceptional clarity. Its originality stems from the creative and highly intuitive synthesis of two fields: it (re-)frames the challenge of Video VAE design through the clear and powerful analogy of video codec standards (I-frame vs. motion residual), which has been overlooked by prior work, offering a path to more accessible and effective VAEs for future video generation models.

**Weaknesses:**

Your paper's stated goal is enhancing VAEs for latent diffusion models, but the experiments almost exclusively validate reconstruction quality, not generative performance. The single, non-comparative generative experiment in the appendix is insufficient. To substantiate its main claim, the paper must add a direct, comparative experiment showing that LDM training on VC-VAE latents is superior to training on baseline latents.

Additionally, the qualitative results are not convincing. In Figure 3, the visual improvements of VC-VAE over other strong baselines are not clear. The paper needs stronger visual evidence to demonstrate that its quantitative gains lead to meaningful perceptual improvements.

Typos:
*“Gererative”* in Figure 7 description
Mulitiple times of *“Visulization”*
Line 050, video codecs standard need space here

**Questions:**

See in Weaknesses.

---

### Note · Authors · 2025-11-12

I have read and agree with the venue's withdrawal policy on behalf of myself and my co-authors.